# Cereal Stem Stress: In Situ Biomechanical Characterization of Stem Elasticity

**D. Jo Heuschele [1,\*], Taina Acevedo Garcia [2], Joan Barreto Ortiz [3] , Kevin P. Smith [1] and Peter Marchetto [4,5]**

[1] Department of Agronomy and Plant Genetics, University of Minnesota, Saint Paul, MN 55108, USA; smith376@umn.edu
[2] Washington Technology Magnet, Saint Paul, MN 55117, USA; gtania817@gmail.com
[3] Department of Horticultural Science, University of Minnesota, Saint Paul, MN 55108, USA; jbarreto@umn.edu
[4] Department of Bioproducts and Biosystems Engineering, University of Minnesota, Saint Paul, MN 55108, USA; petmar@sensinginc.com
[5] Sensing, Roseville, MN 55113, USA
[\*] Correspondence: heus0023@umn.edu; Tel.: +1-612-625-3763

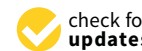

**Featured Application: The featured application of this work is the optimization of breeding techniques for plants susceptible to being damaged by wind.**

**Abstract:** Stem lodging is the bending or breakage of stems in the wind that result in negative economic impacts to producers and processors of small grain crops. To address this issue, plant breeders attempt to quantify lodging using proxy traits such as stem structure and biomechanics. Stem lodging is a function of both stem strength and elasticity. In this paper, we explore the biomechanics of stems approaching the lodging, or permanent bending, condition. Oat, wheat, and two types of barley varying in lodging resistance were exposed to standard growing conditions over the course of a season. Their capability of returning from a bent to unbent state was characterized using a push force meter that measured resistant force and displacement over time. Changes in stem energy and power were then calculated using displacement and force measurements. Lodging susceptibility could be differentiated by stem strength, displacement and change in power measurements depending on small grain species without damaging the plant. These measurements could be used by small cereal grain breeding programs as proxy traits to determine lodging susceptibility without destructively testing or waiting for storm events, thus saving time and resources.

**Keywords:** lodging; stem; plant biomechanics; biophysics; agriculture

## 1. Introduction

Lodging is a complex trait in which both external and internal forces contribute to overall change in plant structural integrity. Plants respond by either bending with the force or remaining rigid. A stem that is too rigid is at risk of becoming unanchored from the substrate via root slippage (root lodging) or buckling somewhere along the stem (stem lodging). Both types of lodging in small grain cereals can decrease yield [1] and grain quality [2] resulting in large economic losses to producers and processors [3]. Plant breeders currently screen for lodging resistance using a visual rating scale to quantify the degree to which lodging occurs after weather events with strong winds. One limitation of this method is that weather events with wind speeds that are either too low or too high will not reveal phenotypic variation for lodging among breeding lines and therefore limit the selection that can be imposed by the breeder [1]. Since weather events are stochastic in nature, for years researchers have been attempting

to use physical mechanics to describe and quantify lodging in cereal grains independent of weather events. Multiple methods have been developed to quantify bending and stem strength in plants. Grafius and Brown [4], developed a method that relied on attaching a load (i.e., chains) to the tip of an oat inflorescence and measuring the curvature of the stem (cLr). Stems that had a greater curvature were correlated with lodging resistance [4]. This method was laborious and thus abandoned for a quicker in situ method called snap score where an individual applied pressure to stems using an arm or stick and then visually rated the amount of immediate recovery with a score value [5]. Although the variance of snap score is similar to the cLr [5], the precision is variable from person to person and practice is required for mastery of the technique. As the field of electronics advanced, in situ pull force [6] and push force meters [7] to measure stalk strength were developed and improved [8,9].

Pull force meters [6] assume stem deformation to be a Hookean spring phenomenon in one dimension. By assuming that the stem behaves as a Hookean spring (i.e., F = −kx applies to it) this model, ignores the nature of tissue deformation in a plant. Additionally, pull force meters specialize in single stem testing and are more likely to damage plants in situ, which makes this tool more suited for measuring stem failure in larger cereal plants such as maize or sorghum. Push force meters differ by measuring the average the force of a group of stems making these instruments ideal for small cereal grains such as wheat, rice and oats. However, the comparison of these measurements requires similar seeding densities between samples. Push force meters use a pivot approach [7,9]; modeling stem strength in this way accounts for tissue deformation in a non-linear elastic phenomenon of two dimensions. The pivot model assumes that motion is rotationally symmetric around the axis of growth; however this type of motion may not be true in all cases because prevailing winds may act to strengthen the growth of the stem in anisotropic ways [10]. This type of modeling, similar to the pull force meters, does not take into account the elasticity of the stem along its length, the conical form of the stem across its height, or the cLr [4].

Both current designs of push and pull force meters allow for quantification of stem strength which has been associated with lodging resistance [6,8,11]. Unfortunately, measuring just stem strength does not take into account elasticity, the ability of the stem to return from deformation in more than one dimension, which plays a role in a stem's ability to recover from wind gusts [4,12,13]. Although the estimation of lodging heritability is improved when using stalk strength compared to the traditional rating scale [7,11], it is clear that the complexity of lodging cannot be explained by stalk strength alone. Understanding the physical modes of action surrounding lodging and being able to quantify them in situ would allow breeders to enhance genetic gain toward lodging-resistant crops providing producers and processors with an improved product. In this paper, we illustrate a detailed biomechanical analysis that can be used to determine resistance or susceptibility to lodging without damaging the plant. We also validate a method to quantify both stem strength and components of elasticity in situ using the Stalker, a push force meter (Sensing, Roseville, MN, USA) [9] originally designed to only quantify stem strength.

## 2. Materials and Methods

### 2.1. Biomechanics

We measured the force applied to a stem, the half-height of the stem (where the force was applied), the angle of the stem, and the time point of each force measurement (i.e., one measurement per milli second) using the Stalker push force meter [9]. Calculations from these data gave the line segment length of displacement of the panicle (the top of the stem), the radius of the line segment (from twice the half-height), the energy applied (and returned), and the power applied in the bending. Please note that E in this paper will represent energy in Joules instead of the Young's modulus.

As seen in Figure 1, the angle of the push force meter (and thence the stem) is increased, then decreased. In doing this at a relatively constant rate (holding ω, the angular velocity, as constant), the force applied to bend a stem, or set of stems, will be uniform and continuous. Mathematically,

there is an inflection point at max, meaning that it is not a continuous function, and therefore not strictly differentiable. Figure 1 is similar to the force curve, seen in Figure 2a, which also shows an inflection point at $F_{max}$, which is isochronous with max. The fact that $F(\theta)$ of the entire series would not pass the vertical line test, and would hence not be a function, necessitates the separation of the two paths as in Figure 2b into bending and unbending.

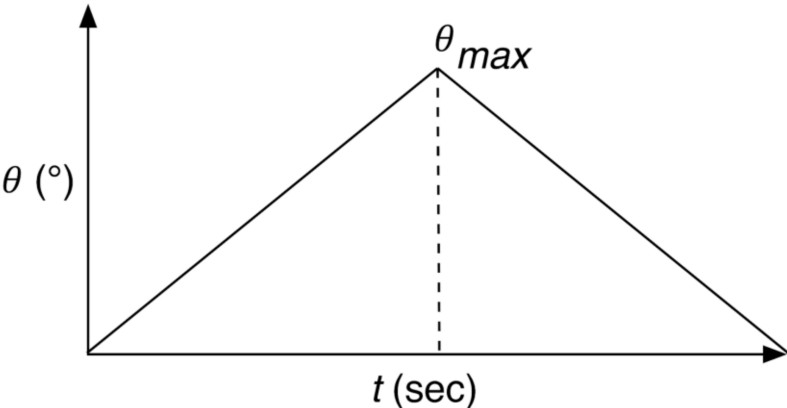

**Figure 1.** Graphical representation of the angle ($\theta$) as a function of time in seconds, as it is recorded in the data logs.

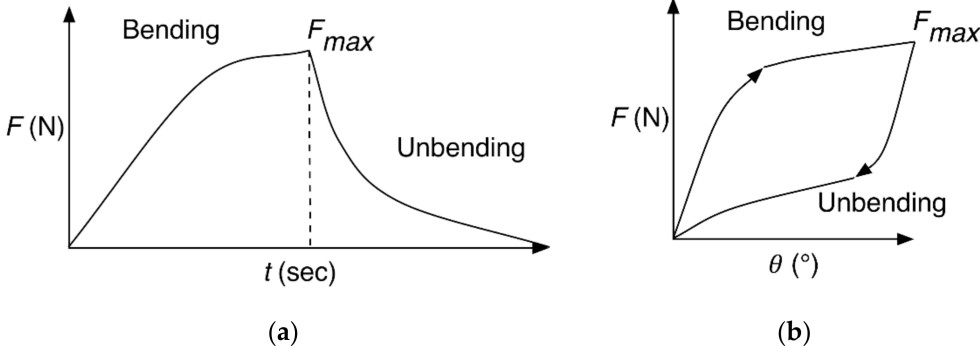

**Figure 2.** (**a**) Force as a function of time, as it is recorded in the data logs; (**b**) Force vs. angle. $F(\theta)$ is not a complete function, therefore it must be decomposed into two functions: the bending function and unbending function.

The bending of a given stem deposits an amount of energy into the pivot point around which it is being bent, as in Figure 3. If this energy is to be calculated, the total path length of the bend ($\ell$) is needed and may be calculated from the angle and the half-height (h) of the stem, as seen in Figure 3.

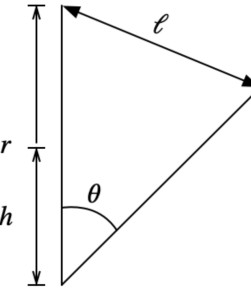

**Figure 3.** An illustration of the bending of a stem of length $r$ to create a line segment of length $\ell$. The half-height, where the measuring device contacts the stem, is a distance $h$ above the ground, and the final angle of bending is denoted as $\theta$.

The path length can be seen in Figure 4 as the independent variable, while the force is the dependent variable. Each of the two functions can be integrated to determine the amount of deposited energy (in the bending curve) and the amount of returned energy (in the unbending curve). The energy differential (ΔE), or the difference between the two integrals, gives the stored potential energy that stays in the stem.

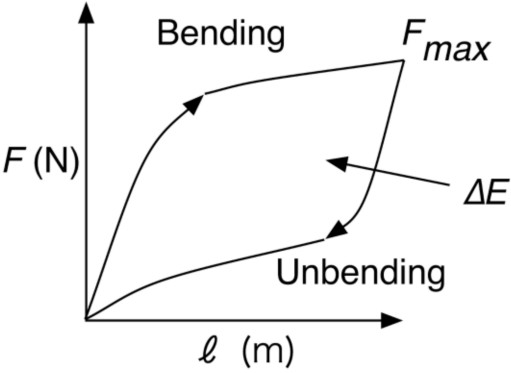

**Figure 4.** After the line segment length has been calculated from the angle, the hysteresis plot of $F_{bending}$ $(\ell)$ and $F_{unbending}(\ell)$ can be created, and the total change in energy can be calculated from the difference of the integrals of the two functions.

In this system, the length of the line segment is used instead of the total arc length because the plants in question do not always bend in a circular path [4]. The energy is measured as the displacement along this line segment as it lengthens to its end (bending to around 45°) and as it returns to vertical (unbending).

To calculate the total energy difference between bending and unbending, the energy of both bending and unbending must first be calculated as:

$$E_{bending} = \int_{O}^{\ell max} F_{bending}(\ell)d\ell = \int_{O}^{\theta max} F_{bending}(2h\ \cos(\theta))d\theta \tag{1}$$

$$E_{unbending} = \int_{O}^{\ell max} F_{unbending}(\ell)d\ell = \int_{O}^{\theta max} F_{unbending}(2h\ \cos(\theta))d\theta \tag{2}$$

where $\ell_{max}$ is the maximum displacement in the line segment, $\ell$ is the line segment displacement, $h$ is the half-height of the stem, is the angle, $\theta_{max}$ is the ultimate angle of the bend, $F_{bending}$ and $F_{unbending}$ are the bending and unbending force data in N, respectively, and $E_{bending}$ and $E_{unbending}$ are the bending and unbending energies in J, respectively.

$$\Delta E = \int_{O}^{\theta max} F_{bending}(2h\ \cos(\theta))d\theta - \int_{O}^{\theta max} F_{unbending}(2h\ \cos(\theta))d\theta \tag{3}$$

where the difference of the two energies is $\Delta E$.

$$P = \frac{E_{bending}}{t_{bending}} - \frac{E_{unbending}}{t_{unbending}} \tag{4}$$

So numerically:

$$\Delta E = \sum_{i=0}^{M} F_{bending} \left(2h\cos(\theta_i)\right)\Delta\theta - \sum_{j=0}^{N} F_{unbending} \left(2h\cos\left(\theta_j\right)\right)\Delta\theta \tag{5}$$

where F(t) → F(n) for a given sampling frequency, *n/t*.

To find the inflection point, the maximum angle must be found, which is correlated with max(F(n)). This inflection point allows the splitting of the time series into the bending and unbending portions. Once this is done, the two functions can be evaluated numerically, and their integrals taken and subtracted.

## 2.2. Plant Material

Four cultivars of four different cereal grains were grown in a randomized complete block design for a total of 16 cultivars: oat, wheat, 2-row and 6-row barley. For the purpose of this study, 2-row and 6-row barley are treated as different cereals even though they are the same species, because of their unique morphological and genetic characteristics [14–17]. For each cereal, two cultivars were classified as lodging susceptible and two were lodging resistant (Table 1), except for wheat. The lodging status was determined using historical data (https://triticeatoolbox.org). Cultivars were planted in single row plots of 3 m in length, 30 cm spacing between rows and a density of 269 seeds/m$^2$; a row of winter wheat was planted in between cereals to reduce weed growth and light competition. Plant rows were tested for strength by pushing the whole plot (row) to 45 degrees at half height, with a push force meter at two weeks after flowering and again at grain maturity. During each time point, plant height was recorded as the distance between the ground and the tip of the inflorescence before each strength measurement. Measuring the height at the time of the strength measurement was important because some lines continued to elongate between flowering and grain maturity. The force meter was calibrated and used as described in Heuschele et al. [9].

**Table 1.** Cultivars tested with their historic lodging classification.

| Cultivar | Cereal Crop | Historic Lodging Classification |
|---|---|---|
| ND-Genesis | 2-row barley | Susceptible |
| AC-Metcalf | 2-row barley | Susceptible |
| Pinnacle | 2-row barley | Resistant |
| Conlon | 2-row barley | Resistant |
| Quest | 6-row barley | Susceptible |
| Celebration | 6-row barley | Susceptible |
| Stellar-ND | 6-row barley | Resistant |
| Tradition | 6-row barley | Resistant |
| Gopher | Oat | Susceptible |
| ND-021052 | Oat | Susceptible |
| IL-8721 | Oat | Resistant |
| Reins | Oat | Resistant |
| MN11394-6 | Wheat | Susceptible |
| Shelly | Wheat | Resistant |
| Rollag | Wheat | Resistant |
| Linkert | Wheat | Resistant |

To verify cultivar lodging type, plants were assessed for lodging incidence and the stem angle in relation to the ground a day after each storm event. During the 2019 field season, there were four significant storm events. For each plant row, the overall lodging angles were calculated by taking the average angle from all storm events. This average was used instead of a single time point at maturity, because depending on the plant's development during a storm event, cultivars may have recovered

to an upright position by harvest. The full experiment was conducted on two separate plantings in St. Paul, MN during 2019.

### 2.3. Data Extraction

The push force meter recorded mechanical load (unitless digital numbers) and angle (degrees) for each millisecond the device was activated at each plot and exported the data files as comma separated values (csv) (Table 2).

**Table 2.** Metadata for Stalker csv output.

| Column 1 | Column 2 | Column 3 |
| --- | --- | --- |
| Time (msec) | Angle (°) | Force (unitless digital number) |

A custom R code (R Core Team, 2014) was created to analyze each csv data file that was collected from each device activation (i.e., plant row). (https://github.com/joanmanbar/StalkMeter). The R script collated all raw data csv files from a trial and added plot number and height to each time point. The mechanical load was converted from arbitrary units to newtons based on machine calibration conducted at the beginning of the field season. From the raw data, force at 45°, displacement (m), change in energy (J), and change in power (W) were calculated for each plot using the theory described above. Calculations were undertaken using the algorithm in Figure 5.

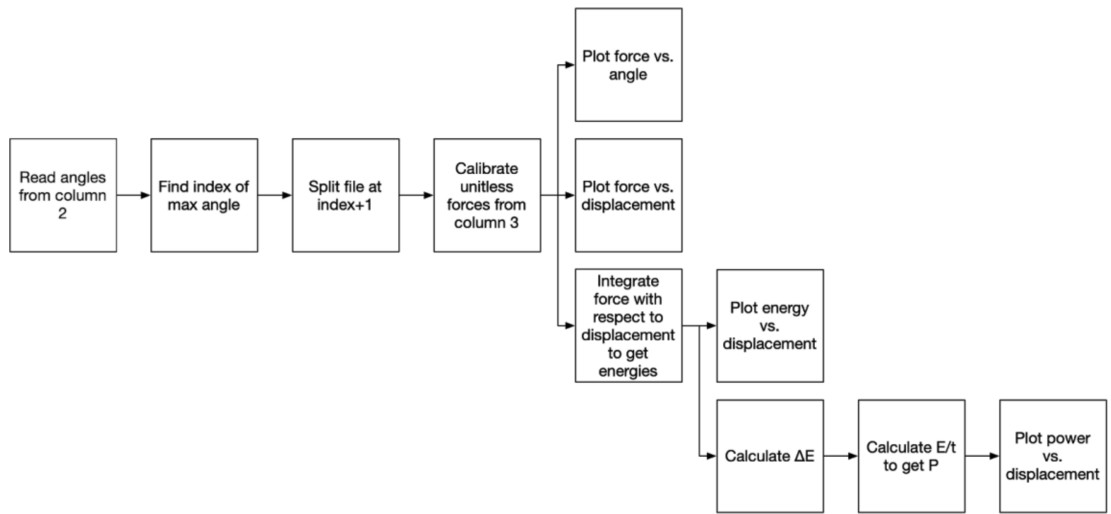

**Figure 5.** Flowchart of the steps taken by the R code during data extraction.

### 2.4. Data Analysis

Mahalanobis distance [18] analysis was conducted to identify and remove outliers from calculated data. On further examination of outliers, we determined them to be a type I systematic instrumentation error, which is a mixture of instrumentation uncertainty and user error. Individual plot data was analyzed using two-way ANOVA to estimate the effect of timing of measurement (factor 1) and lodging category nested within each crop (factor 2). Least square means were then compared with Tukey HSD using JMP Pro 14.2.0 software (SAS Institute Inc., Cary, NC, USA, 1989–2019). All observations were means of eight replications.

## 3. Results

### 3.1. Verification of Lodging Classification

Historically lodging has been rated on a 0–100 or 1–10 scale across multiple environments and years [19]. We measured stem angle which encompassed four storm events, to verify that the cultivars were classified as lodging susceptible and resistant correctly. When all cereals were averaged together by classification, lodging resistant cultivars remained more upright (closer to 90 degrees) than the susceptible cultivars ($p < 0.001$). When data was analyzed by species, oat appears to be the driving the overall angle difference between historical susceptible and resistant cultivars (Figure 6). The lack of difference observed in barley and wheat could be accounted for by the type of storm events. Not all storms create the ideal conditions for lodging events for all cereal species.

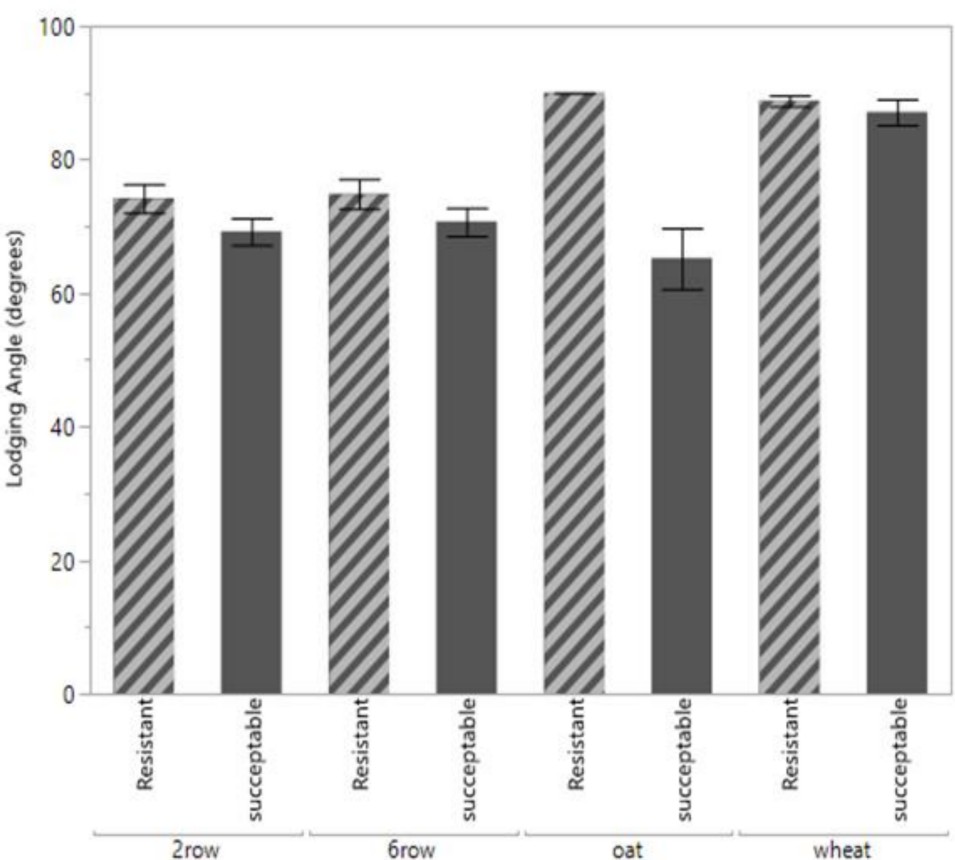

**Figure 6.** Comparison of means for lodging angles of pooled resistant and susceptible lines of all small cereal grains, where the angle of 90 degrees is upright and 0 degrees is perpendicular to the ground. N = 8.

### 3.2. Push Force

Physiological timing of the push force measurement is important to classify lodging. We measured the plants two weeks after flowering and again at grain maturity. These time points represent the beginning and end of the physiological stages where plants are most vulnerable to lodging [1]. At heading, all cereals irrespective of lodging type expressed a similar level of push force required to make the stems bend to 45 degrees. However, at grain maturity, with the exception of 2-row barley, the cereals required less force to be bent at 45 degrees ($R^2 = 0.19$; $p < 0.001$) (Table 3) corresponding with lodging classification. Stems of susceptible cultivars of 6-row barley, oat, and wheat are stronger at heading than at grain maturity, whereas resistant cultivars do not change stem strength over time.

**Table 3.** Crop trait least square means calculated by the push-force meter organized by growth stage and lodging classification nested within the cereal crop.

| Growth Stage | Crop | Lodging Type | Height (m) LS Mean | SE | Force (N) LS Mean | SE | Displacement (m) LS Mean | SE | Energy (J) LS Mean | SE | Power (mW) LS Mean | SE |
|---|---|---|---|---|---|---|---|---|---|---|---|---|
| Heading | 2-row | Resistant | 0.72b [1] | 0.02 | 5.25a | 0.81 | 0.51a | 0.02 | −8.64a | 9.7 | 3.51c | 1.3 |
| | | Susceptible | 0.74b | 0.02 | 6.53a | 0.99 | 0.52a | 0.02 | −40.09b | 11.9 | 5.83b | 1.6 |
| | 6-row | Resistant | 0.75b | 0.04 | 6.51a | 0.85 | 0.53a | 0.02 | −12.25a | 10.6 | 2.60c | 1.4 |
| | | Susceptible | 0.77ab | 0.03 | 7.20a | 0.88 | 0.52a | 0.03 | −10.08a | 10.6 | 4.75b | 1.4 |
| | Oat | Resistant | 0.66c | 0.01 | 7.19a | 0.89 | 0.47b | 0.01 | −14.03a | 10.6 | 5.08b | 1.4 |
| | | Susceptible | 0.78a | 0.03 | 8.44a | 0.85 | 0.53a | 0.02 | −21.90a | 10.1 | 4.03bc | 1.4 |
| | Wheat | Resistant | 0.69b | 0.02 | 6.98a | 0.06 | 0.47b | 0.01 | −4.31a | 7.9 | 3.06c | 1.0 |
| | | Susceptible | 0.71b | 0.01 | 8.20a | 1.15 | 0.48ab | 0.03 | −23.73a | 13.7 | 4.98bc | 1.8 |
| Mature | 2-row | Resistant | 0.73b | 0.02 | 5.00a | 0.81 | 0.50a | 0.02 | −5.43a | 9.7 | 6.53b | 1.3 |
| | | Susceptible | 0.75b | 0.02 | 5.30a | 0.81 | 0.52a | 0.02 | −39.85b | 9.6 | 11.60a | 1.3 |
| | 6-row | Resistant | 0.78a | 0.04 | 5.47a | 0.81 | 0.51a | 0.03 | −5.48a | 9.7 | 7.12b | 1.3 |
| | | Susceptible | 0.78a | 0.03 | 4.70b | 0.81 | 0.52a | 0.02 | −8.56a | 9.7 | 6.69b | 1.3 |
| | Oat | Resistant | 0.68b | 0.02 | 7.29a | 0.85 | 0.44c | 0.02 | −15.08a | 10.1 | 7.76b | 1.4 |
| | | Susceptible | 0.78a | 0.03 | 4.89b | 0.81 | 0.51a | 0.02 | −15.97a | 9.7 | 5.18b | 1.4 |
| | Wheat | Resistant | 0.69b | 0.02 | 6.83a | 0.66 | 0.46b | 0.01 | −25.17a | 7.9 | 8.23b | 1.0 |
| | | Susceptible | 0.72b | 0.01 | 1.87c | 1.15 | 0.48b | 0.01 | −1.95a | 13.70 | 3.56c | 1.8 |

| *F*-values | DF | Height | Force | Displacement | Energy | Power |
|---|---|---|---|---|---|---|
| Growth Stage | 1 | n.s | 18.36 *** | n.s. | n.s. | 17.68 *** |
| Crop | 3 | 4.61 ** [2] | n.s. | 4.86 ** | n.s. | n.s. |
| Lodging [Crop] | 4 | 5.33 ** | 2.86 * | 2.95 * | n.s. | 2.43 * |
| GS*Crop | 3 | n.s. | n.s. | n.s. | n.s. | n.s. |
| GS*L [C] | 4 | n.s. | 4.15 ** | n.s. | n.s. | n.s. |
| Model R$^2$ | 15 | 0.19 ** | 0.21 *** | 0.17 ** | n.s. | 0.23 *** |

[1] Values followed by the same letters indicate within each trait are not significantly different using Tukey HSD ($\alpha$ = 0.05); [2] *p*-value * < 0.05, ** < 0.01, *** < 0.001, n.s. = not significant.

## 3.3. Displacement

The distance that the stem has moved when it reaches a given bending angle (as in Figure 4) is calculated trigonometrically from *h* and θ. Irrespective of plant maturity, displacement differentiates between crops ($p < 0.001$) and lodging susceptibility ($p < 0.01$), where lodging susceptible cultivars were displaced more than resistant cultivars (Table 3) ($R^2 = 0.17$; $p < 0.001$). Specifically, for oat, lodging susceptibility are differentiated by displacement (Table 3). Barley expresses the same displacement distance for both resistant and susceptible cultivars. Some plots were observed to remain at an angle less than 90 degrees after measurement; however, all cultivars were observed to return to an upright position within 24 h of initial measurement.

## 3.4. Energy

Energy was calculated by multiplying the displacement of the row of plants over its bending arc by the amount of force needed to bend the plants. In this way, a total amount of kinetic energy input into the system could be found, and a total amount of kinetic energy taken out of the system during unbending could be calculated as well. The difference between these two energies is the stored (or internal) energy of the stalk (Figure 3 or Figure 4). Overall, changes in energy do not differentiate between plant maturity or lodging type (Table 3); however, 2-row barley lodging resistance could be differentiated. Large variations were found between biological replications of all cereals (Table 3), suggesting the energy measurement can reflect high levels of variability from instrument user to user, as each user's push and pull rates can vary widely making comparisons between trials difficult.

## 3.5. Power

Power was calculated by dividing the kinetic energy calculated above by the amount of time (Joules/second, or Watts) over which it was applied or recovered (Joules/second, or Watts). In this way, the bending and unbending of plants could be standardized for how fast the instrument user pushed on the device, and information on total stem elasticity could be extracted from the collected data. As cultivars matured, the change in power required to bend and unbend lodging resistant genotype stems increased for all cereals (Table 3), except for oat. We observed a change in power between growth stages ($R^2 = 0.23$; $p < 0.001$), but not between crops. At maturity, 2-row barley susceptible cultivars had a greater change in power than the resistant cultivars ($p = 0.0079$). Wheat at maturity had the reverse pattern; lodging resistant cultivars had a greater change in power than susceptible cultivars ($p = 0.0293$).

## 4. Discussion

The purpose of this study was to develop a non-destructive test to reliably predict lodging resistance classification that encompasses both stem strength and elasticity. The original in situ meter, after extensive experimentation, was only capable of determining push force reliably at between 40 and 70 degrees (i.e., estimated stem strength) [7], which has been shown to be predictive of lodging resistance in wheat [7,8] and oat [11], where stronger stems equated to better lodging resistance. The parameters calculated in this study differentiate between historic lodging classifications by accounting for both stem strength and components of elasticity.

### 4.1. Mechanics and Dynamics

A stress strain curve can be applied to plant stems. As seen in Figure 7, the positive linear relationship between stress and strain of a material is known as Young's modulus (*Y*). The yield strength is where the curve becomes non-linear, and the stress begins to relax slightly as the strain increases. Stretching occurs between the yield and ultimate strengths of the material, where the material is taking the most stress that it is physically capable. Stretching then occurs during the elongation (and relaxation) phase, as the material yields even further, breaking bit by bit on the microstructural

level, before its eventual ultimate failure, when a mechanical separation of the material into two or more pieces occurs. Living tissues have a property unique in nature in that they can regenerate their structures and place themselves back in the linear region of the stress strain curve after yield has occurred if active growth of the organism is occurring [20]. In the case of a stem that is being stretched, it elongates as a force bends it past its bend radius, causing the junctions between the cell walls in the outer side of that bend to yield. This separation results in stem snap in plants such as corn and barley [21–24], and can cause the release of roots from the substrate (i.e., soil) resulting in root lodging in oats and other plants [1]. For the purposes of this paper, the measurements of ultimate material failure and root lodging are not discussed.

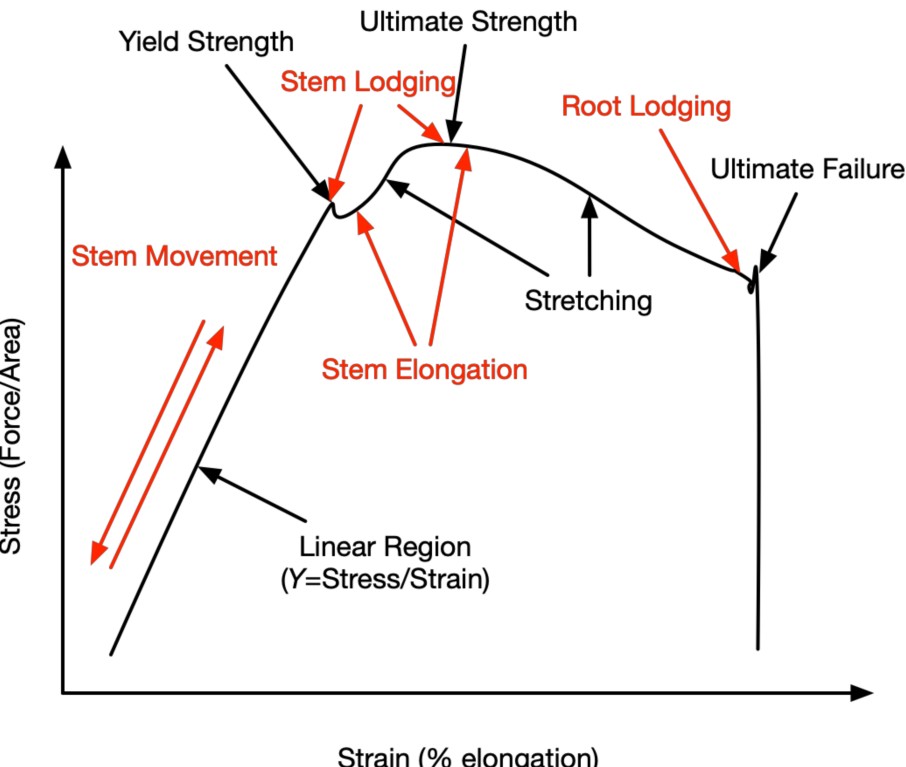

**Figure 7.** Anatomy of a typical stress-strain curve. The text in red indicates movement for a biomaterial, such as the stem or roots of a grass.

Stem strength is defined as being proportional to the amount of force needed to push the stem over to a given angle. The strength of a material can be described in several ways. In the most common definition is yield strength, the amount of force under which the structural integrity of a material begins to fail. There is also the interpretation of strength as the difference between different parts of the stress-strain curve, i.e., when a material begins to deform at a different rate, along with ultimate strength, the maximum stress the material can bear even after deformation. The force on a material in tension is usually denoted as being divided by its cross-sectional area, resulting in the formula:

$$\sigma = \frac{F}{A}$$

where $F$ is the force exerted in Newtons, $A$ is the cross-sectional area in m$^2$, and $\sigma$ the stress in Pascals. The extension or elongation of the material is then calculated as its strain ($\epsilon$), a dimensionless number often expressed as a percentage:

$$\epsilon = \frac{l}{l_{inital}}$$

where $l$ is the current length of the material, $l_{initial}$ is the initial length of the material and is the strain on the material. Thus, the characterization of a material as with a linear relationship between stress and strain (the Young's modulus, $Y$) is only true when the material's stress has not yet transited its yield strength.

The elasticity of the stem is determined by its Young's modulus and yield strength. The elasticity is the property of the stem's strain to remain linear over a wide range of applied forces. Elasticity confers resistance to lodging by allowing for recovery of the stem to close to its initial position, once the force that was acting upon the stem ceases. Thus, smaller septum size [25], and conical stem [23,26] profiles expressed by most cereals would hypothetically confer the most resistance to lodging on a given stem. Therefore, in the case of oat lodging resistance, the difference in displacement measured ($p = 0.0016$) may suggest a difference of overall stem form and septum size. This lack of recovery to the original starting position in lodging susceptible oat cultivars may also suggest a lack of elasticity in the stems or may even be evidence of micro-structural failures within the stem compared to resistant cultivars.

The displacement of the stem under a given force gives an insight to the amount of energy that can be stored in the stem during bending, and that can be recovered during unbending. Biomechanically this means that the material properties of stems that store more energy predispose them to stay bent, rather than recovering by returning the energy in unbending, such that the stored energy stored in the tissues are damaged by fatigue in the bending state.

$$\sum E = PE + KE + U, \; U = TS + \frac{\sigma_{ij}\,\epsilon_{ij}}{2} \tag{6}$$

The total energy in the system is the sum of the mechanical potential and kinetic energies (*PE* and *KE*, respectively) and the internal energy, *U*. The internal energy, in turn, is proportional to the sum of the product of the temperature (*T*) and entropy (*S*), and half of the product of the stresses and strains in the elastic body (the stem) being measured. This equation explains that the remnant portion of the potential energy that is not returned to kinetic energy by swinging the stem back to its initial position is lost as an increase in entropy and temperature in the stem. This internal energy term, *U*, is responsible for the loss in efficiency that creates a difference in bending and unbending energy.

A lower amount of internal energy, and thus more energy returned as kinetic upon unbending, shows that there has been less fatigue damage done to the tissues at the pivot point (i.e., where the stem and root tissue connect). The larger the difference in energy between bending and unbending, the more energy has been invested into the cells of the stem in the form of entropy. This increase in entropy can be due to viscous friction at the pectin interface between ground cells or can be from the weakening of structures in the vascular tissue, as the fibers tear apart [20]. Fatigue is a cascade effect, as the vascular tissues are more brittle than the surrounding ground tissue. As the ground tissue weakens gradually, more and more forces are exerted on the brittle vascular tissue. Finally, this cascade results in a macroscopic structural failure as the fibers underlying the ground tissue fracture, but the ground tissue remains intact, which expresses as buckling in the macro-structure. If the fracture occurs faster than tissue growth, then the plant is susceptible to lodging, while if tissue growth occurs faster, then it is resistant to lodging. The understanding of the set of traits relevant to fracture and flexure is confounded by the fact that in some cases the tissues being characterized may be senescent or dead when measured, therefore stem maturity should be considered when applying measuring force.

### 4.2. Biological Application

Plant structure and physiology change over time; therefore, timing of trait measurements is important. Although growth on a gross spatial–temporal scale appears to be linear and equal across all dimensions, this is not always the case. Cell division only occurs at meristems, and in the case of cereals only at the apical meristem. Cell division also occurs only when cell expansion in a region has reached a specific point [27]. On a macro level cereals plants expand outward (i.e., tiller) before initiation of elongation (i.e., increase in height). This pattern of growth repeats over the life span of the plant. When

injury occurs the plant resources can be transported to the injury location and regenerative growth occurs (i.e., callus or scar tissue) [28].

Weather events might also impact timing of a trait measurement, for example plants grown in high soil moisture have higher water potential [29] which is known to impact the mechanical properties [30]. In this study, stem maturity was found to impact stem strength and change in power of the stems as they were bending and unbending. To quantify the most traits during one measurement, the optimum timing for deployment of this machine is at or right before grain maturity and before senescence.

Push force has already been investigated and used in determining lodging resistance potential of wheat for management [8] and breeding decisions [7]. Stem strength could be used as an individual proxy lodging trait for oat breeding [11] as it is in wheat [7,8]. Furthermore, this technique has been incorporated into predictive lodging models [1,31,32]. Similar to previous studies, wheat classified as lodging susceptible in this experiment had a significant reduction in stem strength. Oat also expressed similar trends, where lodging susceptible lines exhibited less push-force.

Some of the measured components of elasticity could differentiate between lodging resistance and susceptibility depending on the cereal type. In oat, initial displacement differentiated between lodging classifications, whereas the change in energy and power required to bend and unbend a stem differentiated lodging classification of barley and wheat. These differences in classification are most likely linked to the minute differences between the internal structures of the grass species [33,34] and not material properties [35]. The arrangement of internal structures within the stem are more important than their individual material properties [36,37] because the arrangement does not change after initial deformation, unlike material properties [33].

Both stem strength and components of elasticity could allow breeders to make selections within breeding material to increase genetic gain for lodging resistance without waiting for storm events or inducing lodging with expensive environmental equipment. These traits have not been used in plant research for lodging selection as of yet. Although our observations from a small number of genotypes provides some new insight, exploring a larger and more genetically diverse collection of genotypes would yield a better understanding of heritability and genetic architecture underlying these traits. The development of a selection index that combines stalk strength, displacement, and change in power would take into account both strength and elasticity of a stem and could be used to replace the current rating scale. However, this type index also needs to be investigated with a larger genetic pool.

## 5. Conclusions

The biomechanics of the stem system are subtly complex. Various characteristics in the stem, from sizes and material properties of individual cells to anatomical configurations of the several tissues, conspire to create a system whose failure modes are difficult to characterize. Adding to the complexity, stems can heal at different rates that may categorize them as susceptible or resistant to lodging. One of the goals of breeding lodging resistant crops is to increase tolerance to the stress of increased load (i.e., seed weight) leading to stem failure. Thus, an understanding of the mechanism of failure, and early measurements leading to predictions in lodging susceptibility and crop loss will be useful for grain breeding programs concerned with lodging.

In this work we presented a multi-variate method to estimate stem strength and the components of elasticity in situ for plant breeding purposes. Cereals should be characterized for strength and components of elasticity at grain maturity but before senescence. Stalk-strength and changes in power and energy can be used individually or together as proxy traits for lodging resistance; however, more in-depth genetic verification testing should be conducted.

**Author Contributions:** Project conceptualization, P.M. and D.J.H.; software, J.B.O.; validation, D.J.H., T.A.G. and J.B.O.; formal analysis, D.J.H. and T.A.G.; writing—original draft preparation, D.J.H. and P.M.; writing—review and editing, K.P.S.; supervision, K.P.S.; funding acquisition, D.J.H. All authors have read and agreed to the published version of the manuscript.

**Funding:** This research was funded by Minnesota Department of Agriculture grant no. 122130 and University of Minnesota (UMN) Rapid Agricultural Response Fund grant no. AES00RR234. P.M.M.'s work was partially funded by the MNDRIVE RSAM initiative.

**Acknowledgments:** The authors would like to thank Dimitri von Ruckert for the field management, and Daniel Furuta for code maintenance and hardware help.

**Conflicts of Interest:** The authors declare no conflict of interest.

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
