# Peer review of "Cereal Stem Stress: In Situ Biomechanical Characterization of Stem Elasticity"

_applsci, doi:10.3390/app10227965_

Round 1

Reviewer 1 Report

Line 97 what does f() mean?

Line 105 Figure 3 needs to be with its title on the same page

Line 146 m2 change to be m2

Line 161 table 1, in the paragraph before, authors said” For each cereal, two cultivars were 143 classified as lodging susceptible and two were lodging resistant” however, for wheat there are one cultivar Susceptible and three cultivars resistant. One of the statements is not correct!  

Author Response

Comment 1:

Line 97 what does f() mean?

Response:

Theta symbol was missing.  Added theta to line 97.

Comment 2:

Line 105 Figure 3 needs to be with its title on the same page

Response:

Inserted a page break to ensure Figure 3 and the title were on the same page.  This changed the formatting for Table 1, so lines 157-163 were moved to follow Table 1 instead of preceding Table 1.

Comment 3:

Line 146 m2 change to be m2

Response:

On line 148 m2 was changed to a superscript 2.

Comment 4:

Line 161 table 1, in the paragraph before, authors said” For each cereal, two cultivars were 143 classified as lodging susceptible and two were lodging resistant” however, for wheat there are one cultivar Susceptible and three cultivars resistant. One of the statements is not correct!

Response:

Wheat lines provided contained only one lodging susceptible line.  Text (line 146) was edited to reflect a difference in classification for wheat. “classified as lodging susceptible and two were lodging resistant (Table 1), except for wheat.”

Reviewer 2 Report

The topic of the article is very interesting. In researching steam lodginig not only knowledge of stem strength is important but also knowledge of stem elasticity. Authors validate a method to quantify both stem strength and components of elasticity in situ without damaging the plants. This method seems promising and useful for breeders for breeding and selection of crops resistant to stem lodging.

This looks to be a well-conducted study and is well written with just a few minor errors (see below)...

Title: seems appropriate

Introduction: a suitable background to the study and concludes with clear hypotheses, relevant literature cited.

Materials and methods:

The methodology looks to be suitable for such a study.

P 6: Please place the Table 2. right after ...(Table 2)... in the text (L156). This is followed by r code information and Figure 5 after that.

Results :

L188: insert a space: ...year(19).We measured....

Figure 6: Correct the graph (line above is interrupted). Please remove or correct.

Figure 3: Please provide the information about sample size.

Discussion and Conclusions:

Discussion well written and supported by relevant literature. The conclusions are clear and consistent.

Author Response

Comment 1:

P 6: Please place the Table 2. right after ...(Table 2)... in the text (L156). This is followed by r code information and Figure 5 after that.

Response:

Thank you for the suggestion, we originally choose not to place Table 2 directly after the interdiction of the table in the text as it would break up the flow of the paragraph text.  However, due to aesthetics we agree separating Table 2 from Figure 5 makes visualizing the two easier. 

Comment 2:

L188: insert a space: ...year(19).We measured....

Response:

Line 191 we inserted a space between the period and the start of the next sentence.

Comment 3:

Figure 6: Correct the graph (line above is interrupted). Please remove or correct.

Response:

We are unsure as to what line on figure 6 the reviewer is commenting on.  This may have been auto corrected as other edits in formatting where be done.

Comment 4:

Figure 3: Please provide the information about sample size.

Response:

Figure 3 does not reflect any data analysis and therefore would not require sample size information.  However, we did add sample size data to figure 6, “N = 8.”.